

# Counterdiabatic Hamiltonians for multistate Landau–Zener problem

**Kohji Nishimura**[1][⋆] **and Kazutaka Takahashi**[1]

**1** Department of Physics, Tokyo Institute of Technology, Tokyo 152–8551, Japan

⋆ knishimura@stat.phys.titech.ac.jp

## Abstract

We study the Landau–Zener transitions generalized to multistate systems. Based on the work by Sinitsyn et al. [Phys. Rev. Lett. **120**, 190402 (2018)], we introduce the auxiliary Hamiltonians that are interpreted as the counterdiabatic terms. We find that the counterdiabatic Hamiltonians satisfy the zero curvature condition. The general structures of the auxiliary Hamiltonians are studied in detail and the time-evolution operator is evaluated by using a deformation of the integration contour and asymptotic forms of the auxiliary Hamiltonians. For several spin models with transverse field, we calculate the transition probability between the initial and final ground states and find that the method is useful to study nonadiabatic regime.



## 1  Introduction

The Landau–Zener (LZ) formula is an analytic solution for a dynamical two-level quantum system [1,2]. The system Hamiltonian has a linear-time dependence and the time evolution of the state is exactly solvable. By examining together with some generalized results [3,4], we now have a firm picture of nonadiabatic transitions between two instantaneous eigenstates in two-level time-dependent Hamiltonians. This result has a great influence on various fields of physics and we can find a vast amount of applications in literature.

Since the energy-level crossing occurs in general quantum systems, we expect that a similar picture holds for multistate systems where the time-dependent Hamiltonian takes the form

$$\hat{H}(t) = \hat{Z} + t\hat{X}, \tag{1}$$

with arbitrary operators $\hat{Z}$ and $\hat{X}$. Under this Hamiltonian with some minor conditions, the transition probability from the ground state at $t \to -\infty$ to the highest state at $t \to \infty$ was conjectured by Brundobler and Elser (BE) [5]. The following studies revealed that the formula is indeed correct and some extensions are possible [6–12]. The formula can be interpreted as a natural generalization of the original LZ formula.

The same type of the Hamiltonian is used in the method of quantum annealing [13–19]. It is a quantum-mechanical metaheuristic algorithm for solving combinatorial optimization problems. To obtain the optimized solution, we treat Ising Hamiltonians with a transverse field whose time dependence is linear in the standard protocol. Starting from the ground state of the Hamiltonian at $t = -\infty$ where the transverse-field term $\hat{X}$ is dominant, we consider the time evolution until the Ising term $\hat{Z}$ becomes dominant at $t = 0$. Although the standard LZ protocol considers the time evolution until $t$ goes to infinity, we treat the same Schrödinger equation and expect that the LZ picture is still applicable [20, 21]. The field is changed very slowly and the adiabatic picture is applied except around the crossing point. Although this adiabatic approximation gives a reasonable picture, the dynamical description is required to calculate the transition probability, which will be useful to estimate the efficiency of the protocol.

In the optimization problem, we are basically interested in the probability that the system remains in the ground state. Unfortunately, the BE formula cannot be applied to this situation. As we discuss below, the description of the transition from the ground state to the highest state requires few information about the system in contrast to describing the transitions to the other states.

The standard form of the two-level LZ Hamiltonian is written as $\hat{H}(t) = \frac{J}{2}\hat{\sigma}^z + \frac{\gamma t}{2}\hat{\sigma}^x$ where $\hat{\sigma}^z$ and $\hat{\sigma}^x$ are Pauli matrices. Then, it can be shown that the time-evolution operator from $t = -T$ to $t = T$ is decomposed as

$$\hat{U} = \hat{U}_T \hat{U}_0 \hat{U}_{-T}, \tag{2}$$

at $T \to \infty$. $\hat{U}_{\pm T}$ are diagonal in $\hat{\sigma}^x$ basis and are rapidly-oscillating as functions of $T$. We are not interested in these phase contributions and the transition probabilities from eigenstates at $t = -T \to -\infty$ to eigenstates at $t = T \to \infty$ are determined by $\hat{U}_0$ only. $\hat{U}_0$ is independent

of $T$ and is given by

$$\hat{U}_0 = \begin{pmatrix} \sqrt{p} & -\sqrt{1-p} \\ \sqrt{1-p} & \sqrt{p} \end{pmatrix}, \tag{3}$$

where

$$p = \exp\left(-\frac{\pi J^2}{2\gamma}\right). \tag{4}$$

$p$ represents the transition probability from the lower state at $t \to -\infty$ to the higher state at $t \to \infty$ (and vice versa). This is the LZ formula. This decomposition of the unitary operator looks reasonable since the nonadiabatic transition only occurs around $t = 0$. The time domain is decomposed into adiabatic domain $|t| \gg J/\gamma$, which is described by $\hat{U}_{\pm T}$, and diabatic domain $|t| < J/\gamma$, described by $\hat{U}_0$. Then, to calculate the transition probability we may focus only on $\hat{U}_0$. This operator is independent of $T$ and is analytically tractable. It is a fundamental question whether such a decomposition is generally possible and how we can calculate the transition probability between arbitrary states in the general LZ-type Hamiltonian.

Recently, an interesting approach was proposed in Ref. [22]. The Hamiltonian contains several parameters and they are treated as dynamical variables. Using the introduced auxiliary Hamiltonians, we can deform the one-dimensional path $-T \to T$ of the time evolution to a multi-dimensional one in parameter space. The auxiliary Hamiltonian plays a role of the generator for the corresponding parameter. Several solvable models were treated to construct the auxiliary Hamiltonians and to demonstrate that the method is useful to describe the time evolution [22–24].

In this paper, we use the method of the auxiliary Hamiltonians to demonstrate the separation of the unitary operator as in Eq. (2). We also show that the auxiliary Hamiltonians are closely related to the counterdiabatic terms discussed in the method of shortcuts to adiabaticity (STA) [25–30]. By using the auxiliary Hamiltonians, we evaluate the time-evolution operator for several spin models with transverse field to calculate the transition probabilities. Our method is general and is applicable to the general Hamiltonian which is not necessarily solvable. The previous studies treated solvable Hamiltonians and the physical meaning of the auxiliary Hamiltonians was not clear [22–24]. The present work is useful not only to clarify the physical meaning of the auxiliary Hamiltonians but also to calculate the transition probabilities for general time-dependent systems.

The organization of this paper is as follows. In Sec. 2, we study the general structure of time-dependent Hamiltonian. Combining the idea of Ref. [22] with the result from STA, we find a general form of the auxiliary Hamiltonians. The method is applied to the LZ Hamiltonian. In Sec. 3, we demonstrate the decomposition of the time-evolution operator in Eq. (2). Based on the general results, we numerically calculate the transition probability for several spin models in Sec. 4. The final section 5 is devoted to discussions and conclusions.

## 2 Structure of time-dependent Hamiltonian

### 2.1 Auxiliary Hamiltonian for parameter generation

We consider a general time-dependent Hamiltonian $\hat{H}(t, \boldsymbol{\omega})$ with a set of parameters $\boldsymbol{\omega} = (\omega_1, \omega_2, \dots)$. The Schrödinger equation for the state $|\psi(t, \boldsymbol{\omega})\rangle$ is written as

$$i\frac{\partial}{\partial t}|\psi(t, \boldsymbol{\omega})\rangle = \hat{H}(t, \boldsymbol{\omega})|\psi(t, \boldsymbol{\omega})\rangle. \tag{5}$$

In Ref. [22], the derivative with respect to one of the parameters was considered to define the auxiliary Hamiltonian as

$$i\frac{\partial}{\partial\omega_j}|\psi(t,\boldsymbol{\omega})\rangle = \hat{H}_j(t,\boldsymbol{\omega})|\psi(t,\boldsymbol{\omega})\rangle. \tag{6}$$

The auxiliary Hamiltonian $\hat{H}_j(t,\boldsymbol{\omega})$ plays the role of generator for the corresponding parameter $\omega_j$. $\hat{H}$ and $\hat{H}_i$ satisfy the consistency conditions

$$i\frac{\partial\hat{H}_j(t,\boldsymbol{\omega})}{\partial t} - i\frac{\partial\hat{H}(t,\boldsymbol{\omega})}{\partial\omega_j} = [\hat{H}(t,\boldsymbol{\omega}),\hat{H}_j(t,\boldsymbol{\omega})], \tag{7}$$

$$i\frac{\partial\hat{H}_k(t,\boldsymbol{\omega})}{\partial\omega_j} - i\frac{\partial\hat{H}_j(t,\boldsymbol{\omega})}{\partial\omega_k} = [\hat{H}_j(t,\boldsymbol{\omega}),\hat{H}_k(t,\boldsymbol{\omega})]. \tag{8}$$

The equations are interpreted as the zero curvature condition in noncommutative geometry.

It is generally a difficult task to find the auxiliary Hamiltonians for a given $\hat{H}(t,\boldsymbol{\omega})$. In the previous studies [22–24], the condition that the both sides of the above two equations are respectively equal to zero was examined to treat several solvable models. To examine more general cases, we show a relation between $\hat{H}$ and $\hat{H}_j$ by using the method of STA.

A general time-dependent Hamiltonian is written as

$$\hat{H}(t,\boldsymbol{\omega}) = \hat{H}_0(\boldsymbol{\lambda}(t,\boldsymbol{\omega})) + \frac{\partial\boldsymbol{\lambda}(t,\boldsymbol{\omega})}{\partial t}\cdot\hat{\xi}(\boldsymbol{\lambda}(t,\boldsymbol{\omega})), \tag{9}$$

where $\boldsymbol{\lambda}(t,\boldsymbol{\omega}) = (\lambda_1(t,\boldsymbol{\omega}),\lambda_2(t,\boldsymbol{\omega}),\dots)$ is a set of time-dependent parameters. Each term of the Hamiltonian is given respectively by

$$\hat{H}_0(\boldsymbol{\lambda}) = \sum_n \epsilon_n(\boldsymbol{\lambda})|n(\boldsymbol{\lambda})\rangle\langle n(\boldsymbol{\lambda})|, \tag{10}$$

$$\hat{\xi}(\boldsymbol{\lambda}) = i\sum_{m\neq n}|m(\boldsymbol{\lambda})\rangle\langle m(\boldsymbol{\lambda})|\partial_{\boldsymbol{\lambda}}n(\boldsymbol{\lambda})\rangle\langle n(\boldsymbol{\lambda})|. \tag{11}$$

$\epsilon_n(\boldsymbol{\lambda})$ is real and $\{|n(\boldsymbol{\lambda})\rangle\}$ represents a set of orthonormal basis. The solution of the Schrödinger equation is given by the adiabatic state of $\hat{H}_0(\boldsymbol{\lambda})$:

$$|\psi(t,\boldsymbol{\omega})\rangle = \sum_n c_n e^{-i\theta_n(t,\boldsymbol{\omega})}|n(\boldsymbol{\lambda}(t,\boldsymbol{\omega}))\rangle, \tag{12}$$

where $c_n$ is a constant determined from the initial condition at $t = t_0$ and $\theta_n$ is represented by the sum of dynamical phase and geometric phase:

$$\theta_n(t,\boldsymbol{\omega}) = \int_{t_0}^t dt'\,\epsilon_n(\boldsymbol{\lambda}(t',\omega)) - i\int_{\boldsymbol{\lambda}_0}^{\boldsymbol{\lambda}(t,\omega)} d\boldsymbol{\lambda}\cdot\langle n(\boldsymbol{\lambda})|\partial_{\boldsymbol{\lambda}}n(\boldsymbol{\lambda})\rangle. \tag{13}$$

We assume that the initial parameter $\boldsymbol{\lambda}(t_0,\boldsymbol{\omega}) = \boldsymbol{\lambda}_0$ is independent of $\boldsymbol{\omega}$. This separation of the Hamiltonian means that the state is written in terms of the eigenstates of the first term and the second term prevents nonadiabatic transitions. Although it is nontrivial to find this separation for a given total Hamiltonian, the separation is always possible in principle [31]. We also note that the form of $\boldsymbol{\lambda}(t,\boldsymbol{\omega})$ cannot be obtained straightforwardly from the total Hamiltonian since the total Hamiltonian is dependent not only on $\boldsymbol{\lambda}(t,\boldsymbol{\omega})$ but also on the time derivative of $\boldsymbol{\lambda}(t,\boldsymbol{\omega})$.

When the state is evolved under this Hamiltonian, the average energy at each time is determined by $\hat{H}_0$ and the state is changed by the counterdiabatic term $\hat{\xi}$. This means that

the Hamiltonian is separated according to its roles: energy and generator. $\hat{\xi}_j$ corresponds to the generator for the parameter $\omega_j$, which implies a relation to the auxiliary Hamiltonian $\hat{H}_j$. In fact, it is straightforward to show that the counterdiabatic terms satisfy the zero curvature condition

$$i\frac{\partial \hat{\xi}_k(\boldsymbol{\lambda})}{\partial \lambda_j} - i\frac{\partial \hat{\xi}_j(\boldsymbol{\lambda})}{\partial \lambda_k} = [\hat{\xi}_j(\boldsymbol{\lambda}), \hat{\xi}_k(\boldsymbol{\lambda})]. \tag{14}$$

This is obtained from the spectral representation in Eq. (11). We also obtain the form of $\hat{H}_i$ by taking the derivative of Eq. (12) with respect to $\omega_j$ as

$$\hat{H}_j(t, \boldsymbol{\omega}) = \hat{H}_{j0}(t, \boldsymbol{\omega}) + \frac{\partial \boldsymbol{\lambda}(t, \boldsymbol{\omega})}{\partial \omega_j} \cdot \hat{\xi}(\boldsymbol{\lambda}(t, \boldsymbol{\omega})), \tag{15}$$

where

$$\hat{H}_{j0}(t, \boldsymbol{\omega}) = \sum_n \frac{\partial}{\partial \omega_j}\left(\int_{t_0}^t \mathrm{d}t'\, \epsilon_n(\boldsymbol{\lambda}(t', \boldsymbol{\omega}))\right)|n(\boldsymbol{\lambda}(t, \boldsymbol{\omega}))\rangle\langle n(\boldsymbol{\lambda}(t, \boldsymbol{\omega}))|. \tag{16}$$

We note that the structure of the Hamiltonians is insensitive to the choice of $\epsilon_n$ and is mainly determined by the counterdiabatic terms. The simplest choice is to set $\epsilon_n = 0$. Then, the auxiliary Hamiltonians are completely equivalent to the counterdiabatic terms.

## 2.2 Landau–Zener Hamiltonian

In the following we focus on the Hamiltonian

$$\hat{H}(t, \boldsymbol{\omega}) = J\hat{Z} + \gamma t\hat{X}, \tag{17}$$

where $\hat{Z}$ and $\hat{X}$ are arbitrary operators. We take one of the parameters $\gamma$ to be positive without loss of generality. We also assume that the spectrum of the operator $\hat{X}$ is nondegenerate so that we can apply the BE formula to this system [5]. When $\hat{Z}$ and $\hat{X}$ are chosen as two of the Pauli operators, we obtain the standard LZ Hamiltonian. Here our motivation is to set $\hat{Z}$ as an Ising model and $\hat{X}$ as the transverse field.

The Schrödinger equation is written as

$$\frac{i}{\sqrt{\gamma}}\frac{\partial}{\partial t}|\psi(t, \boldsymbol{\omega})\rangle = \left(\frac{Jt}{\sqrt{\gamma}t}\hat{Z} + \sqrt{\gamma}t\hat{X}\right)|\psi(t, \boldsymbol{\omega})\rangle. \tag{18}$$

This equation clearly shows that the state is given by a function of $\sqrt{\gamma}t$ and $Jt$. We set the parameter $\boldsymbol{\lambda}$ as

$$\boldsymbol{\lambda}(t, \boldsymbol{\omega}) = \boldsymbol{\omega}t = (\sqrt{\gamma}t, Jt). \tag{19}$$

The Hamiltonian is decomposed as

$$\hat{H}(t, \boldsymbol{\omega}) = \frac{\partial \boldsymbol{\lambda}(t, \boldsymbol{\omega})}{\partial t} \cdot \hat{\xi}(\boldsymbol{\lambda}(t, \boldsymbol{\omega})) = \sqrt{\gamma}\hat{\xi}_1(\boldsymbol{\lambda}(t, \boldsymbol{\omega})) + J\hat{\xi}_2(\boldsymbol{\lambda}(t, \boldsymbol{\omega})). \tag{20}$$

$\hat{\xi}_1$ and $\hat{\xi}_2$ are related to each other as

$$\hat{\xi}_1(\boldsymbol{\lambda}) = \lambda_1\hat{X} + \frac{\lambda_2}{\lambda_1}\hat{Z} - \frac{\lambda_2}{\lambda_1}\hat{\xi}_2(\boldsymbol{\lambda}). \tag{21}$$

We can write the Schrödinger equation as

$$i\frac{\partial \boldsymbol{\lambda}(t,\boldsymbol{\omega})}{\partial t}\cdot\frac{\partial}{\partial \boldsymbol{\lambda}(t,\boldsymbol{\omega})}|\psi(\boldsymbol{\lambda}(t,\boldsymbol{\omega}))\rangle=\frac{\partial \boldsymbol{\lambda}(t,\boldsymbol{\omega})}{\partial t}\cdot\hat{\xi}(\boldsymbol{\lambda}(t,\boldsymbol{\omega}))|\psi(\boldsymbol{\lambda}(t,\boldsymbol{\omega}))\rangle. \tag{22}$$

We extend this equation so that the equation holds for arbitrary choices of $\boldsymbol{\lambda}(t,\boldsymbol{\omega})$. Then, we obtain

$$i\frac{\partial}{\partial \boldsymbol{\lambda}}|\psi(\boldsymbol{\lambda})\rangle=\hat{\xi}(\boldsymbol{\lambda})|\psi(\boldsymbol{\lambda})\rangle, \tag{23}$$

and the zero curvature condition in Eq. (14). The explicit form of the zero curvature condition with the relation (21) is

$$\left(\lambda_1\frac{\partial}{\partial \lambda_1}+\lambda_2\frac{\partial}{\partial \lambda_2}+1\right)\hat{\xi}_2(\boldsymbol{\lambda})=\hat{Z}-i[\lambda_1^2\hat{X}+\lambda_2\hat{Z},\hat{\xi}_2(\boldsymbol{\lambda})]. \tag{24}$$

We note that $\hat{\xi}$ is not necessarily equal to the counterdiabatic term. Generally, $\hat{\xi}$ includes additional terms that commute with $\hat{H}_0$, just like the first term in Eq. (15). However, the only important point in the following analysis is that $\hat{\xi}$ satisfies the zero curvature condition. Several other remarks are made below: (i). $\hat{\xi}_1$ and $\hat{\xi}_2$ cannot be directly obtained from the original Hamiltonian. These new operators are required when we consider the deformation of the path between initial and final Hamiltonians as we show in the following. (ii). Even if $\hat{H}$ is real symmetric, $\hat{\xi}_{1,2}$, and $\hat{H}_{1,2}$, generally have complex elements. Previous studies only treated the case where the auxiliary Hamiltonians are real symmetric [22–24]. (iii). The auxiliary Hamiltonians in the present example are given by $\hat{H}_j=t\hat{\xi}_j$ with $j=1$ and 2. (iv). $\hat{H}_1$ and $\hat{H}_2$ are not independent. We find this nontrivial property by using STA.

It is convenient to rewrite the differential equation (24) as a function of $\lambda=\lambda_1$ and $g=\lambda_2/\lambda_1$. We put

$$\hat{\xi}_2(\boldsymbol{\lambda})=\frac{1}{\lambda}e^{-i\lambda^2\hat{X}/2}\hat{\xi}(\lambda,g)e^{i\lambda^2\hat{X}/2}. \tag{25}$$

Then, we obtain

$$\frac{\partial}{\partial\lambda}\hat{\xi}(\lambda,g)=\hat{Z}(\lambda)-ig[\hat{Z}(\lambda),\hat{\xi}(\lambda,g)], \tag{26}$$

where

$$\hat{Z}(\lambda)=e^{i\lambda^2\hat{X}/2}\hat{Z}e^{-i\lambda^2\hat{X}/2}. \tag{27}$$

The differential equation is with respect to $\lambda$ and not to $g$. This can be understood from the Schrödinger equation in Eq. (18) where $g$ appears as a time-independent parameter. We also note that the unitary transformation represented by the exponential factor in Eq. (25) is expected from the original form of the Hamiltonian in Eq. (17). Applying the same unitary transformation to the Schrödinger equation, we obtain the transformed Hamiltonian $\hat{\tilde{H}}=J\hat{Z}(\sqrt{\gamma}t)$.

The equation can be formally expressed as

$$\hat{\xi}(\lambda,g)=\hat{V}(\lambda,g)\left[\int_0^\lambda d\tau\,\hat{V}^\dagger(\tau,g)\hat{Z}(\tau)\hat{V}(\tau,g)\right]\hat{V}^\dagger(\lambda,g), \tag{28}$$

with the use of the path-ordered unitary operator

$$\hat{V}(\lambda,g)=\mathrm{P}\exp\left(-ig\int_0^\lambda d\tau\,\hat{Z}(\tau)\right). \tag{29}$$

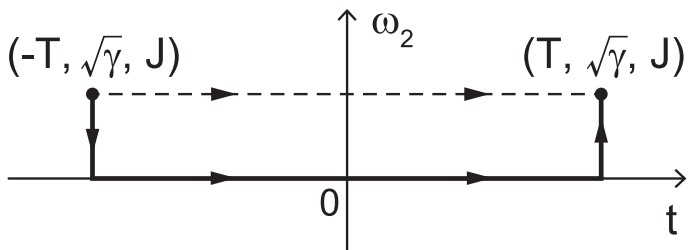

Figure 1: The original path denoted by the dashed line is deformed to the bold line. $\omega_1 = \sqrt{\gamma}$ is kept constant.

To avoid the divergence of $\hat{\xi}_2(\lambda)$ for $\lambda \to 0$, we require the condition $\hat{\xi}(0, g) = 0$, which determines the constant of integration. We can also write a form expanded in terms of $g$:

$$
\begin{aligned}
\hat{\xi}(\lambda, g) = & \int_0^\lambda d\tau \, \hat{Z}(\tau) + \sum_{k=1}^\infty (-ig)^k \int_0^\lambda d\tau_k \int_0^{\tau_k} d\tau_{k-1} \cdots \int_0^{\tau_1} d\tau_0 \\
& \times [\hat{Z}(\tau_k), [\hat{Z}(\tau_{k-1}), \cdots, [\hat{Z}(\tau_1), \hat{Z}(\tau_0)]] \cdots].
\end{aligned}
\tag{30}
$$

## 3 Decomposition of the time-evolution operator

### 3.1 Path deformation

By introducing the auxiliary Hamiltonians, we can consider the path deformation of the time evolution. The unitary time-evolution operator represented by the time-ordered products

$$
\hat{U} = \mathrm{T} \exp\left(-i \int_{-T}^T dt \, \hat{H}(t, \boldsymbol{\omega})\right),
\tag{31}
$$

is equivalent to the path-ordered operator

$$
\hat{U} = \mathrm{P} \exp\left(-i \sum_\mu \int_C d\omega_\mu \hat{H}_\mu(t, \boldsymbol{\omega})\right),
\tag{32}
$$

under the zero curvature condition in Eqs. (7) and (8). Here, we write $\omega_\mu = (t, \omega_1, \omega_2)$, $\hat{H}_\mu = (\hat{H}, \hat{H}_1, \hat{H}_2)$, and C represents a path in $(t, \boldsymbol{\omega})$-space. We fix the initial point $(t, \boldsymbol{\omega}) = (-T, \sqrt{\gamma}, J)$ and final point $(T, \sqrt{\gamma}, J)$ and take the limit of $T \to \infty$.

The path C can be chosen arbitrarily. Here we take the path

$$
(-T, \sqrt{\gamma}, J) \to (-T, \sqrt{\gamma}, 0) \to (T, \sqrt{\gamma}, 0) \to (T, \sqrt{\gamma}, J),
\tag{33}
$$

where each segment represents the straight line, as we show in Fig. 1. The advantage of using this path is that the time variable takes a large constant value at the first and last segments, $|t| = T$, and the operator from the second segment gives a trivial contribution. We have

$$
\begin{aligned}
\hat{U} = & \ \mathrm{P} \exp\left(-iT \int_0^J d\omega_2 \, \hat{\xi}_2(\sqrt{\gamma}T, \omega_2 T)\right) \exp\left(-i \int_{-T}^T dt \, \gamma t \hat{X}\right) \\
& \times \mathrm{P} \exp\left(iT \int_J^0 d\omega_2 \, \hat{\xi}_2(-\sqrt{\gamma}T, -\omega_2 T)\right).
\end{aligned}
\tag{34}
$$

The second exponential factor comes from the second segment and gives the identity operator. By using Eq. (25) and using the property $\hat{\xi}_2(\lambda_1, \lambda_2) = \hat{\xi}_2(-\lambda_1, \lambda_2)$, we can write

$$\hat{U} = e^{-i\gamma T^2 \hat{X}/2} \mathrm{P} \exp\left(-i \int_{-\delta}^{\delta} \mathrm{d}g\, \hat{\xi}(\sqrt{\gamma}T, g)\right) e^{i\gamma T^2 \hat{X}/2}, \tag{35}$$

where $\delta = J/\sqrt{\gamma}$. Then, we only need to know the asymptotic behavior of $\hat{\xi}(\lambda, g)$ at $\lambda \to \infty$.

## 3.2 Asymptotic form

As we can understand from Eq. (28), $\hat{\xi}$ diverges linearly in $\lambda$ at most. The linear divergence is obtained when $\hat{Z}$ has diagonal elements in $\hat{X}$ basis. We also generally find that $\hat{\xi}$ has a logarithmic-divergent term.

We write $\hat{Z} = \hat{Z}_0 + \hat{Z}_1$ where $\hat{Z}_0$ represents the diagonal part and $\hat{Z}_1$ offdiagonal part. At zeroth order in $g$, the diagonal part of $\hat{\xi}$ is linear in $\lambda$. The $j$th element is given by

$$\int_0^\lambda \mathrm{d}\tau\, (\hat{Z}(\tau))_{jj} = \lambda(\hat{Z}_0)_{jj}. \tag{36}$$

The logarithmic divergence comes from the diagonal part at the first order in $g$. The $j$th element is given by

$$
\begin{aligned}
&-ig \int_0^\lambda \mathrm{d}\tau_1 \int_0^{\tau_1} \mathrm{d}\tau_0 [\hat{Z}_1(\tau_1), \hat{Z}_1(\tau_0)]_{jj} \\
=\ & 2g \sum_{k(\neq j)} \int_0^\lambda \mathrm{d}\tau_1 \int_0^{\tau_1} \mathrm{d}\tau_0 \sin\left[\frac{(\tau_1^2 - \tau_0^2)(X_j - X_k)}{2}\right] (\hat{Z}_1)_{jk}(\hat{Z}_1)_{kj} \\
\sim\ & 2g \sum_{k(\neq j)} \frac{(\hat{Z}_1)_{jk}(\hat{Z}_1)_{kj}}{X_j - X_k} \ln\lambda,
\end{aligned}
\tag{37}
$$

where $X_j$ represents the eigenvalue of $\hat{X}$. The estimation of the integral is given in Appendix A.

It should be noticed that these divergent terms are obtained from the dynamical phase calculated in the adiabatic approximation [8]

$$\phi_j(T) = \int^T \mathrm{d}t\, \epsilon_j(t). \tag{38}$$

The instantaneous energy $\epsilon_j(t)$ is evaluated perturbatively and we obtain up to second order in $J$ as

$$\epsilon_j(t) \sim \gamma t X_j + J(\hat{Z}_0)_{jj} + \frac{J^2}{\gamma} \sum_{k(\neq j)} \frac{(\hat{Z}_1)_{jk}(\hat{Z}_1)_{kj}}{(X_j - X_k)t}. \tag{39}$$

Then, we have

$$\phi_j(T) \sim \frac{\gamma T^2 X_j}{2} + J T(\hat{Z}_0)_{jj} + \frac{J^2}{\gamma} \sum_{k(\neq j)} \frac{(\hat{Z}_1)_{jk}(\hat{Z}_1)_{kj}}{(X_j - X_k)} \ln T. \tag{40}$$

The first term corresponds to the phase in Eq. (25), the second to Eq. (36), and the third to Eq. (37). The key observation in Ref. [8] is that these are only divergent contributions to the phase at $T \to \infty$. In Appendices A and B, we show that the same property holds for $\hat{\xi}$. The divergent terms in $\hat{\xi}$ come from the diagonal part as we estimated above. In the following, we show that they can be represented as a phase in the time-evolution operator and do not give any contribution to the transition probabilities.

### 3.3 Gauge transformation

The separation of the phase can be done by the gauge transformation. We insert the identity operator $\hat{I} = e^{-i\hat{\phi}(\lambda,g)} e^{i\hat{\phi}(\lambda,g)}$ at each slice of the Trotter decomposition in Eq. (35). We assume that $\hat{\phi}$ is diagonal in $\hat{X}$ basis. Then, $\hat{\xi}$ is replaced by the gauge-transformed operator

$$\hat{\tilde{\xi}}(\lambda,g) = e^{i\hat{\phi}(\lambda,g)} \left( \hat{\xi}(\lambda,g) - \partial_g \hat{\phi}(\lambda,g) \right) e^{-i\hat{\phi}(\lambda,g)}. \tag{41}$$

Using the arbitrariness of $\hat{\phi}$, we can remove the divergent contribution from the diagonal part of $\hat{\tilde{\xi}}$ as

$$\hat{\tilde{\xi}}_0(\lambda,g) = \hat{\xi}_0(\lambda,g) - \partial_g \hat{\phi}(\lambda,g), \tag{42}$$

$$\hat{\tilde{\xi}}_1(\lambda,g) = e^{-i\hat{\phi}(\lambda,g)} \hat{\xi}_1(\lambda,g) e^{-i\hat{\phi}(\lambda,g)}, \tag{43}$$

where the subscript 0 denotes the diagonal part and 1 denotes the offdiagonal part. It is possible to choose $\hat{\phi}$ such that the condition $\hat{\tilde{\xi}}_0 = 0$ is satisfied. The addition of phase in $\hat{\tilde{\xi}}_1$ does not induce any additional divergence. Thus, we conclude that the time-evolution operator is decomposed as

$$\hat{U} = e^{-i\frac{\gamma T^2}{2}\hat{X} - i\hat{\phi}(\sqrt{\gamma}T,\delta)} \hat{U}_0(\delta) e^{i\frac{\gamma T^2}{2}\hat{X} + i\hat{\phi}(\sqrt{\gamma}T,-\delta)}, \tag{44}$$

$$\hat{U}_0(\delta) = \mathrm{P}\exp\left( -i \int_{-\delta}^{\delta} dg\, \hat{\tilde{\xi}}(\infty,g) \right), \tag{45}$$

as is expected in Eq. (2). To calculate transition probabilities between the eigenstates of $\hat{X}$, we may focus on $\hat{U}_0$ which is only dependent on $\delta = J/\sqrt{\gamma}$ and contains no divergent contributions.

## 4 Transition probability

It is still a difficult problem to find the explicit form of $\hat{\xi}$ in general systems. Even if we can find $\hat{\xi}$, we have to manage the path-ordered integral in Eq. (45). Although these calculations look a formidable task, the BE formula implies that the unitary operator is well-approximated by $\hat{\xi}$ at zeroth order in $g$:

$$\hat{U}_0(\delta) \sim \exp\left( -2i\delta \int_0^{\infty} d\tau\, \hat{Z}_1(\tau) \right). \tag{46}$$

The formula treats the transition probability from the lowest state at $t \to -\infty$ to the highest state at $t \to \infty$ (and vice versa). Their states are represented by the same eigenstate of $\hat{X}$ and are denoted by $|0\rangle$. The transition probability is evaluated by the second-order cumulant expansion as

$$|\langle 0|\hat{U}_0(\delta)|0\rangle|^2 \sim \exp\left[ -4\delta^2 \int_0^{\infty} d\tau_1 \int_0^{\infty} d\tau_2 \langle 0|\hat{Z}_1(\tau_2)\hat{Z}_1(\tau_1)|0\rangle \right]$$

$$= \exp\left[ -2\pi\delta^2 \sum_{k(\neq 0)} \frac{\langle 0|\hat{Z}_1|k\rangle\langle k|\hat{Z}_1|0\rangle}{X_0 - X_k} \right], \tag{47}$$

where $X_0$ represents the eigenvalue of $\hat{X}$ for $|0\rangle$. This is exactly the BE formula [5]. We note that the derivation of Eq. (47) is not rigorous since this equation is derived by applying small-$\delta$ approximation. It is nontrivial that Eq. (47) holds for arbitrary values of $\delta$.

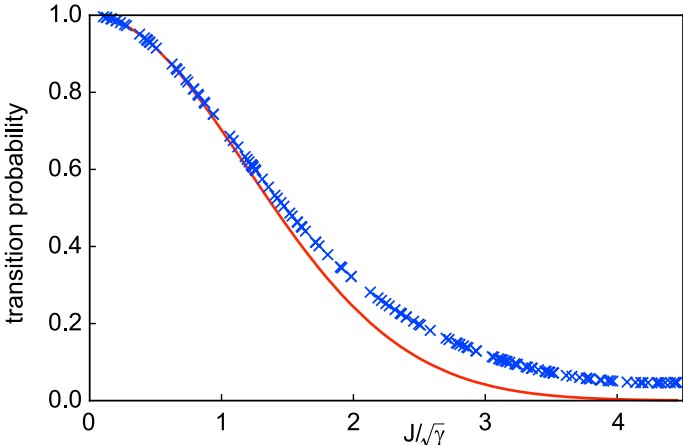

Figure 2: The BE formula in Eq. (47) (red solid line) and the zeroth-order approximation (×, blue) in Eq. (46) for $\hat{H}_{2-\text{body}}$ in Eq. (48). We set $N = 10$.

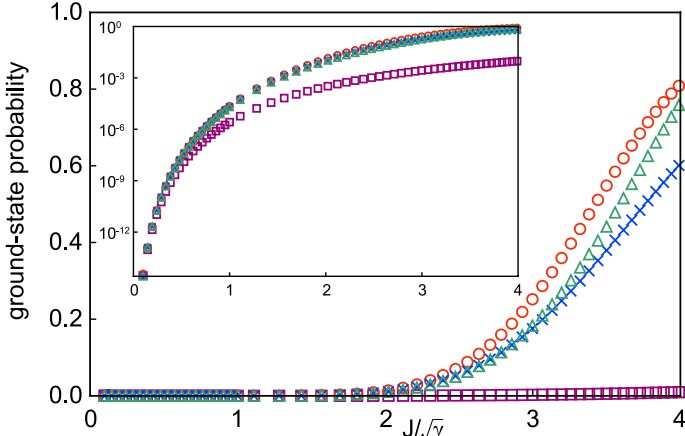

Figure 3: Ground-state probability. Plotted are numerical calculation for $\hat{H}_{2-\text{body}}$ with $\hat{Z}_0$ in Eq. (48) (□, purple), numerical calculation for $\hat{H}_{2-\text{body},0}$ without $\hat{Z}_0$ in Eq. (52) (◯, red), zeroth order approximation in Eq. (46) (×, blue), and estimation by Eq. (53) (△, green). We set $N = 10$. (Inset) Logarithmic scale.

In most of the applications we are interested in the transition probability between the ground states at $t \to -\infty$ and at $t \to \infty$. Below, for several spin models, we study the transition probabilities by using the truncation approximation of $\hat{\tilde{\xi}}(\infty, g)$ keeping the lower-order terms in $g$. The approximation is valid for nonadiabatic regime where $J < \sqrt{\gamma}$ and can be a complementary method of the adiabatic approximation.

## 4.1 Two-body interacting spin model

First, we study the two-body interacting Ising model with the transverse field:

$$\hat{H}_{2-\text{body}}(t, \boldsymbol{\omega}) = \frac{J}{N} \hat{S}_z^2 + \gamma t \hat{S}_x. \tag{48}$$

$\hat{\boldsymbol{s}} = (\hat{S}_x, \hat{S}_y, \hat{S}_z)$ represents the total spin of the spin-1/2 system $\hat{\boldsymbol{s}} = \frac{1}{2} \sum_{i=1}^{N} \hat{\boldsymbol{\sigma}}_i$ where $N$ represents the number of 1/2-spin. The Hamiltonian commutes with $\hat{\boldsymbol{s}}^2$ and we consider the block

with the quantum number

$$\hat{\boldsymbol{S}}^2 = \frac{N}{2}\left(\frac{N}{2}+1\right). \tag{49}$$

The ground state and the highest state are contained in this block and the Hamiltonian is represented by a $(N+1) \times (N+1)$ matrix. We start the time evolution from the ground state of the initial Hamiltonian and examine the transition probabilities at large $T$. In this model, the BE formula is evaluated as

$$|\langle 0|\hat{U}_0(\delta)|0\rangle|^2 = \exp\left[-\frac{\pi\delta^2}{8}\left(1-\frac{1}{N}\right)\right]. \tag{50}$$

We note that this Hamiltonian has diagonal elements of $\hat{Z}$ in $\hat{X}$ basis. In the present model, the decomposition of $\hat{Z} = \hat{Z}_0 + \hat{Z}_1$ is written as

$$\hat{S}_z^2 = \frac{1}{2}\left(\hat{\boldsymbol{S}}^2 - \hat{S}_x^2\right) + \frac{1}{2}\left(\hat{S}_z^2 - \hat{S}_y^2\right). \tag{51}$$

To examine the effect by the diagonal term, we also treat a modified Hamiltonian with the diagonal part of $\hat{Z}$ removed:

$$\hat{H}_{2-\text{body},0}(t,\boldsymbol{\omega}) = \frac{J}{2N}\left(\hat{S}_z^2 - \hat{S}_y^2\right) + \gamma t\hat{S}_x. \tag{52}$$

We note that the BE formula in Eq. (50) is unchanged by this modification.

We use two kinds of approximation. The first one is to keep only the zeroth-order term in Eq. (30). The time-evolution operator is approximated as Eq. (46). The second approximation is to use some ansatz taking into account the higher-order corrections. We put

$$\hat{U}_0(\delta) = \exp\left(-2i\delta\hat{\bar{\xi}}_0 + [f(\delta)\hat{S}_+^2 + f^*(\delta)\hat{S}_-^2, [\hat{S}_+^2, \hat{S}_-^2]]\right), \tag{53}$$

where $\hat{\bar{\xi}}_0$ represents the zeroth-order term as in Eq. (46) and $\hat{S}_\pm = \hat{S}_y \pm i\hat{S}_z$. The operator form of the double-commutator term is taken from the second term of the expansion in Eq. (30) with some modifications by the gauge transformation. Instead of taking into account the path-ordered integral, we put

$$f(\delta) = a\delta^3 + b\delta^5, \tag{54}$$

and complex parameters $a$ and $b$ are determined by the method of least squares so that Eq. (53) reproduces the BE formula. Then, using the determined parameters, we numerically calculate the transition probability that the state remains in the ground state.

The zeroth-order approximation of the transition probability from the initial ground state to the final highest state is plotted in Fig. 2. We see that the approximation gives a reasonable description of the BE formula at small $\delta = J/\sqrt{\gamma}$. We find a small deviation at large $\delta$. In Fig. 3, we plot the transition probability to the ground state. The numerical calculations are compared with the results of the zeroth-order approximation and of the estimation by Eq. (53). We see that the numerical result for Eq. (52) is well described by theoretical estimations. A large deviation for Eq. (48) is considered to be the presence of the diagonal elements of $\hat{Z}_0$. Since the diagonal components do not contribute to the BE formula, it is impossible to estimate the undetermined parameters in the present method. In any case, the theoretical analysis works well in the nonadiabatic regime.

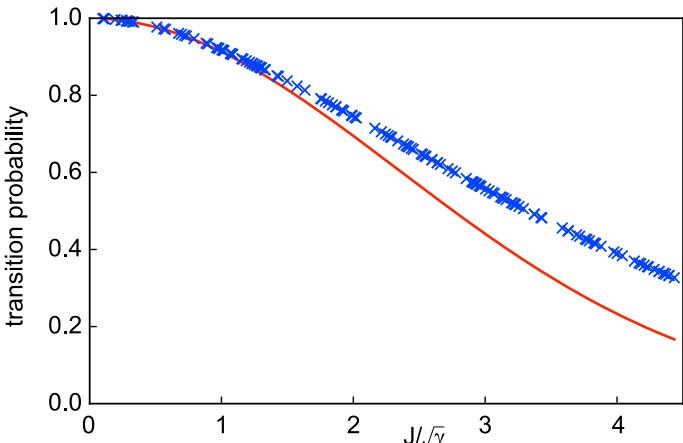

Figure 4: The BE formula in Eq. (47) (red solid line) and the zeroth-order approximation (×, blue) for $\hat{H}_{3-\text{body}}$ in Eq. (55). We set $N = 10$.

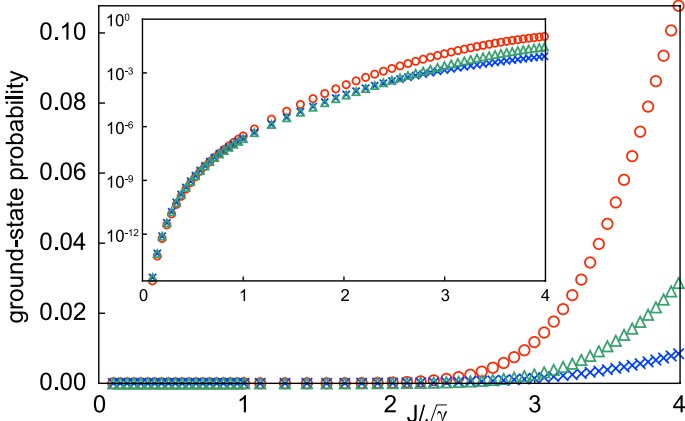

Figure 5: Ground-state probability for $\hat{H}_{3-\text{body}}$ in Eq. (55) with $N = 10$. Legend is the same as in Fig. 3. (Inset) Logarithmic scale.

## 4.2 Three-body interacting spin model

For comparison, we study the three-body interacting Hamiltonian

$$\hat{H}_{3-\text{body}}(t, \boldsymbol{\omega}) = \frac{J}{N^2}\hat{S}_z^3 + \gamma t \hat{S}_x. \tag{55}$$

We note that $\hat{Z}_0 = 0$ in this case. The BE formula is evaluated as

$$|\langle 0|\hat{U}_0(\delta)|0\rangle|^2 = \exp\left[-\frac{\pi\delta^2}{16N}\left(\frac{11}{2} - \frac{9}{N} + \frac{4}{N^2}\right)\right]. \tag{56}$$

This probability goes to unity at $N \to \infty$. Then, compared to the two-body case, the ground-state probability takes a considerably smaller value. The comparison of the BE formula and the zeroth-order approximation is shown in Fig. 4. The ground-state probability is plotted in Fig. 5. We again find deviations at large $\delta$, basically the same conclusion as the two-body case.

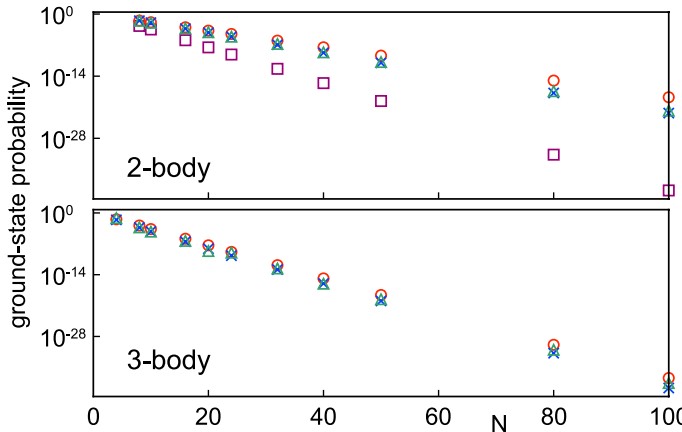

Figure 6: Size dependence of the ground-state probability for $\hat{H}_{2-\text{body}}$ and $\hat{H}_{2-\text{body},0}$ (top) and for $\hat{H}_{3-\text{body}}$ (bottom). We take $\delta = 2.0$. Legend is the same as in Fig. 3.

### 4.3 Size dependence

The BE formula shows that the size dependence of the probability to the highest state is totally different between the two- and three-body systems. We find that, at least for nonadiabatic regime, the probability to the ground state has the scaling Prob $\sim \exp(-Nc)$, as we show in Fig. 6. This scaling is expected to be changed in the superadiabatic regime $\delta \gg 1$ where the static picture holds. In the adiabatic approximation, the transition probability is determined by the energy gap between the ground state and the first-excited state [21]. The energy gap behaves differently between the two- and three-body systems as their models have different kinds of quantum phase transitions. In our numerical calculation, $\delta$ takes smaller values and we find that the scaling is unchanged in the present systems. The same scaling holds for probabilities to the other states except the highest state, which shows that the BE formula is specific to the highest state and does not describe a typical behavior.

### 4.4 Probability distribution

Finally, we examine the transition-probability distribution among the energy levels. The numerical results are shown in Fig. 7. For nonadiabatic regime $\delta \sim 1$, we find that, independently of the models, the transitions to higher levels are dominant. For adiabatic regime $\delta \gg 1$, the behavior is dependent of the models as we expect from the BE formula. We also see that our approximations work well in all the energy levels.

## 5 Conclusion

In conclusion, we have studied the multiple LZ transitions by using the method of the auxiliary Hamiltonians. We find that the auxiliary Hamiltonians introduced in Ref. [22] are equivalent to the counterdiabatic terms in the method of STA and are directly related to the original Hamiltonian. By specifying the time-dependent parameters, we derive the differential equations for the counterdiabatic Hamiltonians. Using the counterdiabatic Hamiltonians and the deformation of the integration contour, we evaluated the time-evolution operator and calculated the transition probabilities. It is instructive to see that the general Hamiltonian contains several counterdiabatic terms.

Even though the present study focused only on the LZ Hamiltonian, our method is general

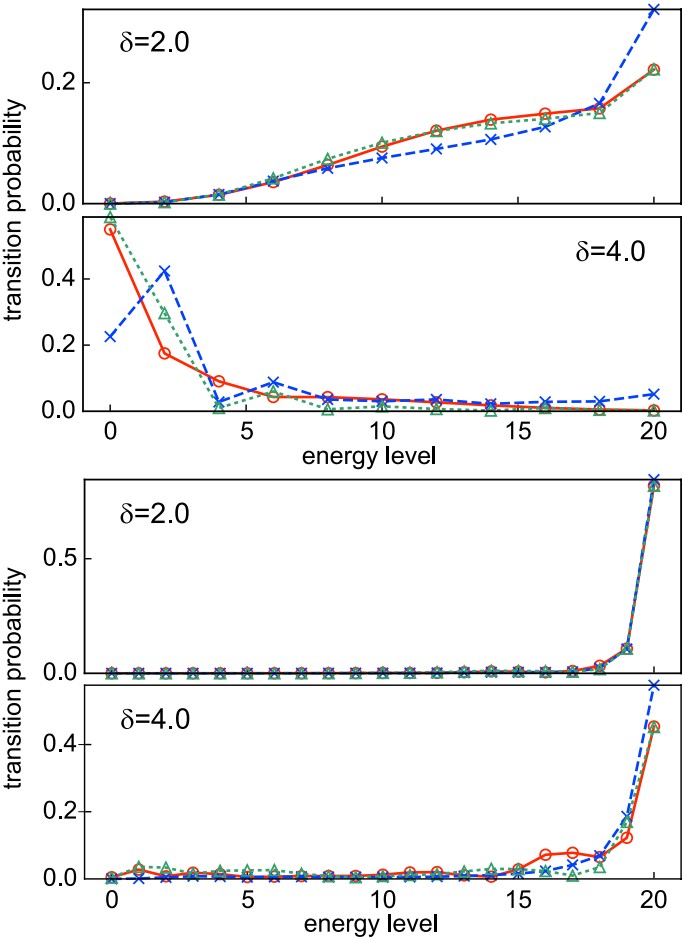

Figure 7: (Top) Transition-probability distributions for $\hat{H}_{2-\text{body},0}$ with $N = 20$. The energy level takes even value: $0, 2, 4, \ldots, 20$. (Bottom) Distributions for $\hat{H}_{3-\text{body}}$ with $N = 20$. The energy level takes integer value: $0, 1, 2, \ldots, 20$. Legend is the same as in Fig. 3.

and is applicable to any Hamiltonians. The spectral representation of the counterdiabatic term in Eq. (11) is not so useful for complicated systems and we can use the differential equation such as Eq. (26) instead. To solve the equation, we can use some familiar methods such as perturbative expansions as we demonstrated in the present work.

As for the problem of the LZ transitions, our method becomes a generalization of the BE formula. The BE formula is applicable to a specific state only and it is hard to generalize the formula for different states. We treat the time-evolution operator, which allows us to estimate transition probabilities between arbitrary eigenstates. Although it is generally difficult to solve the differential equation in our method, a simple approximation gives a reasonable description in the nonadiabatic regime, which becomes a complementary method of the adiabatic approximation.

It is an interesting problem to apply the present method to more general systems. In the present study, we treated the following conditions for simplicity: (i). nondegenerate eigenvalues of $\hat{X}$, and (ii). vanishing of diagonal elements of $\hat{Z}$ in $\hat{X}$ basis. We note the condition (i) is satisfied for the standard transverse-field Ising models. Although the constraint (ii) does not hold in general, this problem can be circumvented by changing the basis. We can also apply our method to other dynamical systems with nonlinear time-dependent terms. Thus,

our method is general enough and various applications can be found, which will be useful to understand general dynamical systems.

## Acknowledgments

We performed all our numerical calculations with the Quantum Toolbox in Python (QuTiP) [32].

**Funding information** This work was supported by JSPS KAKENHI Grant Number JP18J13685 (K.N.) and Number JP26400385 (K.T.).

## A   Two-level system

We study the two-level system. By calculating the asymptotic form of $\hat{\bar{\xi}}(\lambda, g)$ at $\lambda \to \infty$, we demonstrate the decomposition of the time-evolution operator in Eq. (2).

In the two-dimensional case, we only have three independent operators except the identity operator. They are represented by the Pauli operators:

$$\hat{X} = \frac{1}{2}\hat{\sigma}^x, \quad \hat{Y} = \frac{1}{2}\hat{\sigma}^y, \quad \hat{Z} = \frac{1}{2}\hat{\sigma}^z. \tag{57}$$

We can also put

$$\hat{\bar{\xi}}(\lambda, g) = \alpha(\lambda, g)\hat{X} + \beta(\lambda, g)\hat{Y} + \gamma(\lambda, g)\hat{Z}, \tag{58}$$

to write the differential equation (26) as

$$\partial_\lambda \alpha(\lambda, g) = -g\beta(\lambda, g)\cos\frac{\lambda^2}{2} + g\gamma(\lambda, g)\sin\frac{\lambda^2}{2}, \tag{59}$$

$$\partial_\lambda \beta(\lambda, g) = \sin\frac{\lambda^2}{2} + g\alpha(\lambda, g)\cos\frac{\lambda^2}{2}, \tag{60}$$

$$\partial_\lambda \gamma(\lambda, g) = \cos\frac{\lambda^2}{2} - g\alpha(\lambda, g)\sin\frac{\lambda^2}{2}. \tag{61}$$

Using these equations, we can derive the differential equation for $\alpha$ as

$$\left( \frac{1}{\lambda^2}\partial_\lambda^3 - \frac{1}{\lambda^3}\partial_\lambda^2 + \partial_\lambda + \frac{g^2}{\lambda^2}\partial_\lambda - \frac{g^2}{\lambda^3} \right) \alpha(\lambda, g) = \frac{g}{\lambda}. \tag{62}$$

This is a linear nonhomogeneous differential equation and the solution is obtained by combining the particular solution and the general solutions of the corresponding homogeneous equation. We are only interested in the asymptotic form and the solution is basically expressed in power series of $1/\lambda$.

The particular solution is expanded in terms of $g$ and we have

$$\alpha(\lambda, g) = g\left( \ln\lambda + \frac{3}{4\lambda^4} + \cdots \right) + g^3\left( \frac{1-2\ln\lambda}{4\lambda^2} + \cdots \right) + \cdots. \tag{63}$$

The general solution of the homogeneous equation has nonoscillating terms and oscillating ones:

$$\alpha_h(\lambda, g) = gc(g)\left( 1 - \frac{g^2}{2\lambda^2} + \cdots \right) - \frac{gh(g)}{\lambda}\cos\left( \frac{\lambda^2}{2} + \frac{g^2}{2}\ln\lambda + \theta(g) \right) + \cdots. \tag{64}$$

$c(g)$, $h(g)$, and $\theta(g)$ represent undetermined functions. Combining these results and neglecting irrelevant terms, we obtain

$$\alpha(\lambda, g) \sim g \ln \lambda + g c(g) - \frac{g h(g)}{\lambda} \cos\left( \frac{\lambda^2}{2} + \frac{g^2}{2} \ln \lambda + \theta(g) \right). \tag{65}$$

Since the differential equation is in terms of $\lambda$, not of $g$, and only the asymptotic form is obtained in the present analysis, $c_0(g)$, $h(g)$, and $\theta(g)$ are left undetermined. As we can understand from the differential equation, $\alpha(\lambda, g)$ is an odd function in $g$, which shows that $c_0(g)$, $h(g)$, and $\theta(g)$ are even functions of $g$. We also see that this result shows that Eq. (37) holds in the two-level case. Since the integral in Eq. (37) is independent of the operator $\hat{Z}$, the equation holds in the general case as well.

Although the last term in Eq. (65) is negligible at the asymptotic limit, this term is important to find $\beta$ and $\gamma$. By using Eq. (59), we conclude that

$$\beta(\lambda, g) \sim -h(g) \sin\left( \frac{g^2}{2} \ln \lambda + \theta(g) \right), \tag{66}$$

$$\gamma(\lambda, g) \sim h(g) \cos\left( \frac{g^2}{2} \ln \lambda + \theta(g) \right). \tag{67}$$

These are even functions of $g$. As we mentioned in the main body of the present paper, the divergence of $\hat{\xi}$ is logarithmic in $\lambda$ and is contained only in the diagonal part. The log term in the trigonometric functions of the offdiagonal part can be removed by the gauge transformation as we show below.

Now that we have obtained the asymptotic form of $\hat{\xi}$, we make a gauge transformation represented by

$$\hat{\phi}(\lambda, g) = \left( \frac{g^2}{2} \ln \lambda + \int dg\, g c(g) \right) \hat{X}. \tag{68}$$

Then, the gauge-transformed operator is given by

$$\hat{\tilde{\xi}}(\lambda, g) = \tilde{\beta}(g) \hat{Y} + \hat{\gamma}(g) \hat{Z}, \tag{69}$$

where

$$\tilde{\beta}(g) = -h(g) \sin\left( \theta(g) - \int dg\, g c(g) \right), \tag{70}$$

$$\tilde{\gamma}(g) = h(g) \cos\left( \theta(g) - \int dg\, g c(g) \right). \tag{71}$$

This operator is independent of $\lambda$, which means that the divergent contribution can be removed by the gauge transformation. The time-evolution operator is decomposed as

$$\begin{aligned} \hat{U} &= \exp\left[ -i \left( \frac{\gamma T^2}{2} \hat{X} + \hat{\phi}(\sqrt{\gamma} T, \delta) \right) \right] \mathrm{P} \exp\left[ -i \int_{-\delta}^{\delta} dg\, \left( \tilde{\beta}(g) \hat{Y} + \tilde{\gamma}(g) \hat{Z} \right) \right] \\ &\quad \times \exp\left[ i \left( \frac{\gamma T^2}{2} \hat{X} + \hat{\phi}(\sqrt{\gamma} T, -\delta) \right) \right], \end{aligned} \tag{72}$$

where $\delta = J/\sqrt{\gamma}$. Furthermore, the second line can be rewritten as

$$\mathrm{P} \exp\left[ -i \int_{-\delta}^{\delta} dg\, \left( \tilde{\beta}(g) \hat{Y} + \tilde{\gamma}(g) \hat{Z} \right) \right] = e^{i\varphi(\delta)\hat{X}} e^{-if(\delta)\hat{Z}} e^{-i\varphi(\delta)\hat{X}}. \tag{73}$$

This relation can be shown by using that $\tilde{\beta}(g)$ and $\tilde{\gamma}(g)$ are even functions of $g$. Now we have the decomposition in Eq. (2), as it should be. It is not necessary to show this decomposition in the present system because we know the exact solution. However, we see below that this demonstration is useful to apply the method to general systems.

## B   General properties of the auxiliary Hamiltonians

In Appendix A, we showed that the logarithmic divergence appears only in the diagonal part of $\hat{\xi}$ at first order in $g$. We can show that this property holds in general systems. For simplicity, here we only consider the case that the diagonal part of $\hat{Z}$ is zero: $\hat{Z}_0 = 0$.

The general form of the diagonal part of $\hat{\xi}$ at first order in $g$ is written in Eq. (37) and has a logarithmic contribution. The divergence is suppressed in the offdiagonal part of the same term because the integrand is more oscillating. We note that, to find this property, it is required that the spectrum of $\hat{X}$ is nondegenerate. The same consideration is applied to the higher-order terms in $g$. At second order and third order, the diagonal part is respectively written as

$$(-ig)^2 \int_0^\lambda d\tau_2 \int_0^{\tau_2} d\tau_1 \int_0^{\tau_1} d\tau_0 [\hat{Z}_1(\tau_2),[\hat{Z}_1(\tau_1),\hat{Z}_1(\tau_0)]]_{jj}$$

$$= (-ig)^2 \sum_{k,\ell(\neq j)} \int_0^\lambda d\tau_2 \int_0^{\tau_2} d\tau_1 \int_0^{\tau_1} d\tau_0 \, e^{\frac{i}{2}\tau_2^2 X_{jk}+\frac{i}{2}\tau_1^2 X_{k\ell}+\frac{i}{2}\tau_0^2 X_{\ell j}}$$

$$\times (\hat{Z}_1)_{jk}(\hat{Z}_1)_{k\ell}(\hat{Z}_1)_{\ell j} + \cdots, \tag{74}$$

$$(-ig)^3 \int_0^\lambda d\tau_3 \int_0^{\tau_3} d\tau_2 \int_0^{\tau_2} d\tau_1 \int_0^{\tau_1} d\tau_0 [\hat{Z}_1(\tau_3),[\hat{Z}_1(\tau_2),[\hat{Z}_1(\tau_1),\hat{Z}_1(\tau_0)]]]_{jj}$$

$$= (-ig)^3 \sum_{k,\ell,m(\neq j)} \int_0^\lambda d\tau_3 \int_0^{\tau_3} d\tau_2 \int_0^{\tau_2} d\tau_1 \int_0^{\tau_1} d\tau_0 \, e^{\frac{i}{2}\tau_3^2 X_{jk}+\frac{i}{2}\tau_2^2 X_{k\ell}+\frac{i}{2}\tau_1^2 X_{\ell m}+\frac{i}{2}\tau_0^2 X_{mj}}$$

$$\times (\hat{Z}_1)_{jk}(\hat{Z}_1)_{k\ell}(\hat{Z}_1)_{\ell m}(\hat{Z}_1)_{mj} + \cdots, \tag{75}$$

where $X_{jk} = X_j - X_k$. Each term is oscillating rapidly and the integration basically gives a finite contribution. A possibly-divergent contribution arises for a specific choice of the subscript indices so that some cancellation occurs in the phase. For example, by choosing $(j,k,\ell,m) = (j,k,j,k)$ with $j \neq k$ in Eq. (75), we have

$$(-ig)^3 \sum_{k(\neq j)} \int_0^\lambda d\tau_3 \int_0^{\tau_3} d\tau_2 \int_0^{\tau_2} d\tau_1 \int_0^{\tau_1} d\tau_0 \, e^{\frac{i}{2}(\tau_3^2-\tau_2^2)X_{jk}+\frac{i}{2}(\tau_1^2-\tau_0^2)X_{jk}}$$

$$\times (\hat{Z}_1)_{jk}(\hat{Z}_1)_{kj}(\hat{Z}_1)_{jk}(\hat{Z}_1)_{kj} + \cdots$$

$$= (-ig)^3 \sum_{k(\neq j)} \int_0^\lambda d\tau_3 \int_0^{\tau_3} d\tau_2 \int_0^{\tau_2} d\tau_1 \int_0^{\tau_1} d\tau_0 \, 4i \cos\left[(\tau_3^2-\tau_2^2)\frac{X_{jk}}{2}\right]$$

$$\times \sin\left[(\tau_1^2-\tau_0^2)\frac{X_{jk}}{2}\right]((\hat{Z}_1)_{jk}(\hat{Z}_1)_{kj})^2. \tag{76}$$

The integral is the same as that appears in two-level systems. We have already shown that this integral does not give any divergent contribution. We can generalize this consideration to the higher-order terms. Thus, we conclude that, in general systems, the divergence appears only in the diagonal part at first order in $g$.

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
