# Peer review of "Counterdiabatic Hamiltonians for multistate Landau-Zener problem"

_SciPost Physics, doi:SciPost Phys. 5, 029 (2018)_

## Round 1 · Referee Report · Anonymous (Referee 1) · 2018-7-17

Strengths

  1. Technical correctness
  2. Making interesting connections between close research areas and posing some interesting questions for further study
  3. Good presentation
  4. Language

Weaknesses

  1. The approach used is not very original.
  2. This work presents a rather small incremental progress.
  3. Partial list of references

Report

The authors employ the method of auxiliary Hamiltonians for studying multiple Landau-Zener transitions. Among other results, they prove an equivalence between such Hamiltonians and the counterdiabatic terms appearing in the family of methods known as shortcuts to adiabaticity. Although I believe that the main ideas behind this work were already presented in references [19]-[22], the paper is very clear, scientifically rigorous and well-written. In fact, it really excels in these parameters. For that, I am glad to recommend its acceptance upon the following: 1. Since the novelty here is not very clear, I'd suggest the authors to denote it explicitly and possibly think of other means to enhance the main message in comparison to previous works in this area. 2. In the introduction the authors write: "The same type of the Hamiltonian is used in the method of quantum annealing [13–17]". This sentence is very general and not entirely precise. I'd encourage the authors to elaborate more on this topic. 3. With respect to the latter point, but also in general, I'd like to recommend the authors to use "Quantum Spin Glasses, Annealing and Computation" by Tanaka, Tamura and Chakrabarti (e.g. Chs. 5-7). Additional references to the literature would also be appreciated.

Requested changes

  1. Since the novelty here is not very clear, I'd suggest the authors to denote it explicitly and possibly think of other means to enhance the main message in comparison to previous works in this area.
  2. In the introduction the authors write: "The same type of the Hamiltonian is used in the method of quantum annealing [13–17]". This sentence is very general and not entirely precise. I'd encourage the authors to elaborate more on this topic.
  3. With respect to the latter point, but also in general, I'd like to recommend the authors to use "Quantum Spin Glasses, Annealing and Computation" by Tanaka, Tamura and Chakrabarti (e.g. Chs. 5-7). Additional references to the literature would also be appreciated.

---

## Round 1 · Referee Report · Anonymous (Referee 2) · 2018-7-30

Strengths

  1. The paper contains a set of very instructive calculations, which might also be useful for people from other fields.

  2. The presentation is rigorous and clear.

  3. The language (English) probably needs a revision.

Weaknesses

  1. The content lacks novelty. There is no result that adds to the general scenario known already.

Report

In this work, the authors study the multi-level version of the Landau Zener problem in the light of short-cut to adiabaticity. An exact expression for a counter-adiabatic term has been calculated, and it has been shown that the counter-adiabatic Hamiltonians satisfy the zero curvature condition.

The work, though not particularly novel, contains nice and rigorous calculations that might be useful in other contexts. I hence recommend publication of the work in SciPost.

Requested changes

  1. The language has to be revised thoroughly.

  • validity: high
  • significance: good
  • originality: low
  • clarity: high
  • formatting: good
  • grammar: below threshold

---

## Round 1 · Referee Report · Anonymous (Referee 3) · 2018-7-31

Strengths

  1. Analyses are clear.
  2. Presentation is good.
  3. The result is generic and potentially useful.

Weaknesses

  1. The main claim is not clear.
  2. The statement with respect to Eq. (45) is in question. See Report below.

Report

The Landau-Zener problem is a well-known problem in a two-level system. It was solved exactly and applied widely in chemistry as well as physics. The generalization of the Landau-Zener problem has drawn a lot of interests so far. The authors of this manuscript analyzed multi-level Landau-Zener problems using the method of auxiliary Hamiltonians developed by Sinitsyn et al., and gave approximate but potentially useful formulas to compute the transition probabilities, including the transition from the ground state to ground state, for a generic Landau-Zener type of time dependent Hamiltonian with multi levels.

I would like to give one comment and one question on this manuscript as follows.

Comment. The claim of this manuscript sounds vague. To my understanding, the main point of this manuscript should be the invention of an approximate computation method for the transition probability between the initial and final ground states in a generic multi-level Landau-Zener-type model. However, the manuscript stresses auxiliary Hamiltonians rather than transition probabilities. I suggest that Introduction should be revised so that the letter is stressed more.

Question. It is stated that Eq. (45) is exactly the same as the Brundobler-Elser (BE) formula in Sec. 4. This statement is surprising and mysterious to me. This is because, while the BE formula is exact, Equation (45) is obtained after the zeroth order approximation in g of Z_1(tau) and the second-order cumulant expansion. The statement implies that, although the approximation used here should be valid for small g and delta, the resultant formula is exact for any g and delta. Why does this happen? I would like to ask the authors to solve this mystery.

Apart from the above comment and question, this manuscript is technically sound. The analysis is sufficiently clear. After a minor revision, this manuscript will deserve to be accepted for publication.

Requested changes

  1. Revise Introduction so as to stress the approximate computation method for the transition probability between the initial and final ground states in a generic multi-level Landau-Zener-type model.

  2. Add an explanation to resolve the question on Eq. (45).

---

## Round 2 · Author Response

Reply to the referees:

We first would like to thank the referees for reviewing our manuscript. We appreciate their useful comments and recommendations. We have modified the manuscript following the comments. We do hope this revised version deserves publication in SciPost. Below is our response for the comments.

Report 1:

Strengths: 1. Technical correctness

We have found a slightly incorrect description in Section 2.2. $\xi$ introduced in this section is not necessarily equal to the counterdiabatic term. It possibly includes additional contributions. However, the equation to be solved in the following analysis, the zero curvature condition, is unchanged. Therefore, this correction does not affect our conclusion. We also mention that the present analysis is not necessarily restricted to the real symmetric Hamiltonian.

Since the novelty here is not very clear, I'd suggest the authors to denote it explicitly and possibly think of other means to enhance the main message in comparison to previous works in this area.

The main claim of this paper is that the auxiliary Hamiltonians introduced by Sinitsyn et al. (reference [22] in the new manuscript) can be interpreted as the counterdiabatic terms of Shortcuts to Adiabaticity. While the introduction of the auxiliary Hamiltonians is artificial and the physical meaning of these Hamiltonians is unclear in the previous study, we have succeeded in giving a natural interpretation of these Hamiltonians by identifying them as the counterdiabatic terms. Our method can be applied to general Landau-Zener Hamiltonians ($\hat{H} = J\hat{Z}+\gamma t \hat{X}$, where $Z$ and $X$ are arbitrary Hermitian operators), hence it enables us to calculate transition probabilities approximately even if the Hamiltonian is not solvable. Moreover, we may possibly generalize our discussion to Hamiltonians which are not limited to the present Landau-Zener type.

Following the referee's comment, we have modified the introduction to stress the novelty of the present study.

  1. In the introduction the authors write: "The same type of the Hamiltonian is used in the method of quantum annealing [13–17]". This sentence is very general and not entirely precise. I'd encourage the authors to elaborate more on this topic.

  2. With respect to the latter point, but also in general, I'd like to recommend the authors to use "Quantum Spin Glasses, Annealing and Computation" by Tanaka, Tamura and Chakrabarti (e.g. Chs. 5-7). Additional references to the literature would also be appreciated.

Thanks for the comment. When performing quantum annealing, one uses the Hamiltonian $H(t) = J\hat{Z}+\Gamma(t) \hat{X}$ where the operator $Z$ represents the problem of the combinatorial optimization and the operator $X$ is a transverse magnetic field. The parameter $\Gamma(t)$ is decreased slowly in the course of the time evolution to obtain the optimal solution. If we consider the schedule which is linear in $t$, $\Gamma(t) = \gamma t$, and consider the time evolution from $t=-\infty$ to $t=\infty$, the protocol is the same as Landau-Zener protocol. Due to the similarity of these two protocols, we expect that Landau-Zener picture is applicable to analysing the performance of quantum annealing, which was actually discussed in literatures such as Refs.[20, 21] in the new manuscript.

Following the referee's suggestion, we have added a brief introduction to quantum annealing and several references ([17], [19] and [20] in the new manuscript).

Report 2:

Weaknesses: The content lacks novelty. There is no result that adds to the general scenario known already.

Thanks for the recommendation and the comment. We think our work is of importance due to the following three reasons.

First, no previous studies exist that consider the relation between the auxiliary Hamiltonians introduced by Sinitsyn et al. (reference [22] in the new manuscript) and the counterdiabatic terms of Shortcuts to Adiabaticity. We have suggested that the auxiliary Hamiltonians can be interpreted as the counterdiabatic terms. Accordingly, we have succeeded in giving a natural interpretation to the auxiliary Hamiltonians. In the previous study, the auxiliary Hamiltonians were introduced artificially, whose physical meaning was unclear.

Second, our theory can be applied to the general Landau-Zener Hamiltonians which are not necessarily solvable. The previous studies ([22-24] in the new manuscript) treated solvable Hamiltonians. We can develop a systematic approximation method to treat the transition probabilities of the nonsolvable Hamiltonian, as discussed in Section 4.

Finally, our formulation may possibly be applicable to the Hamiltonian which is not limited to Landau-Zener type (e.g. the Hamiltonian $\hat{H} = J\hat{Z}+\gamma f(t) \hat{X}$ with arbitrary time dependence $f(t)$) since we can discuss the correspondence of the auxiliary Hamiltonians to the counterdiabatic terms for arbitrary Hamiltonians, in principle.

We have added some explanations to the manuscript so that the novelty of the present study becomes clear.

Strengths: 2. The presentation is rigorous and clear.

As stated in the reply to the report 1, we have modified an incorrect description in our manuscript, which does not affect our conclusion.

Requested changes: 1. The language has to be revised thoroughly.

Thanks for the advice. We have checked the manuscript thoroughly to improve the presentation.

Report 3:

Strengths: 1. Analyses are clear.

As stated in the reply to the report 1 and report 2, we have modified an incorrect description in our manuscript, which does not affect our conclusion.

Comment. The claim of this manuscript sounds vague. To my understanding, the main point of this manuscript should be the invention of an approximate computation method for the transition probability between the initial and final ground states in a generic multi-level Landau-Zener-type model. However, the manuscript stresses auxiliary Hamiltonians rather than transition probabilities. I suggest that Introduction should be revised so that the letter is stressed more.

Thanks for the useful comment. The main claim we would like to stress in this paper is the correspondence between the auxiliary Hamiltonians and the counterdiabatic terms appeared in Shortcuts to Adiabaticity. However, as the referee pointed out, we consider that a systematic calculation of the transition probabilities is as important as the above claim. Our theory is not only for a formal discussion but also for a practical use.

According to the request, we have modified the introduction to stress the importance of the second point.

Question. It is stated that Eq. (45) is exactly the same as the Brundobler-Elser (BE) formula in Sec. 4. This statement is surprising and mysterious to me. This is because, while the BE formula is exact, Eq. (45) is obtained after the zeroth order approximation in g of $Z_1(\tau)$ and the second-order cumulant expansion. The statement implies that, although the approximation used here should be valid for small $g$ and $\delta$, the resultant formula is exact for any $g$ and $\delta$. Why does this happen? I would like to ask the authors to solve this mystery.

This is a highly nontrivial result and we have not been able to explain why this happens. We suppose that this behavior comes from the fact that the exponent of the BE formula depends only on the factor $\delta^2 (=J^2/\gamma)$ and does not depend on other degrees of $\delta$. The exact mechanism cannot be explained in our series expansion method and a nonperturbative approach would be needed to answer this question completely.

We have added a comment below Eq. (47) in the new manuscript.

---

## Round 2 · List of Changes

1. Modified fifth (paragraph of Eq.(4)) and seventh paragraphs in the introduction to stress the novelty of the present study.
  2. Added a brief introduction to quantum annealing in the third paragraph of the introduction.
  3. Added references [17], [19], [20] on quantum annealing.
  4. Added a sentence below Eq. (45) on the nontriviality of the equation.
  5. Modified the second and third paragraphs in section 2.2.
  6. Presentation improved, and grammatical errors and typos corrected.

---

## Editorial Decision

published